



# Seasonality of archaeal lipid flux and GDGT-based thermometry in sinking particles of high latitude oceans: Fram Strait (79° N) and Antarctic Polar Front (50° S)

Eunmi Park[1,2,3], Jens Hefter[1], Gerhard Fischer[2,3], Morten H. Iversen[1,2], Simon Ramondenc[1,2], Eva-Maria Nöthig[1], Gesine Mollenhauer[1,2,3]

[1]Alfred-Wegener-Institute, Helmholtz-Center for Polar and Marine Sciences, D-27570 Bremerhaven, Germany
[2]MARUM Centre for Marine Environmental Sciences, University of Bremen, D-28334 Bremen, Germany
[3]Department of Geosciences, University of Bremen, D-28334 Bremen, Germany

*Correspondence to*: Eunmi Park (eunmi.park@awi.de)

**Abstract** The relative abundance of individual archaeal membrane lipids, namely of glycerol dialkyl glycerol tetraethers (GDGTs) with the different number of cyclopentane rings, varies with temperature, which enabled their use as paleotemperature proxy. The first GDGT-based index in marine sediments called $TEX_{86}$ is believed to reflect mean annual sea surface temperature (maSST). The $TEX_{86}^L$ is an alternative temperature proxy for ´low´ temperature regions (<15 ℃), where the original $TEX_{86}$ proxy suffers from scattering in a linear calibration with SSTs. However, $TEX_{86}^L$-derived temperatures still display anomalous estimates in polar regions. In order to elucidate the potential cause of the disagreement between $TEX_{86}^L$ estimate and SST, we analyzed GDGT fluxes and $TEX_{86}^L$-derived temperatures in sinking particles collected with time-series sediment traps in high northern and southern latitude regions. At 1296 m depth in the eastern Fram Strait (79° N), a combination of various transporting mechanisms for GDGTs might result in seasonally different sinking velocities for particles carrying these lipids, resulting in strong variability in the $TEX_{86}^L$ signal. The similarity of flux weighted $TEX_{86}^L$ temperatures from sinking particles and surface sediments implies an export of GDGTs without alteration during transport in the Fram Strait. The estimated temperatures correspond to temperatures in water depths of 30−80 m, where nitrification might occur, indicating the favorable depth habitat of Thaumarchaeota. In the Antarctic Polar Front of the Atlantic sector (50° S), $TEX_{86}^L$-derived temperatures displayed warm and cold biases compared to satellite-derived SSTs at 614 m depth, and its flux-weighted mean signal differs from the deep signal at 3196 m. $TEX_{86}^L$-derived temperatures at 3196 m depth and the surface sediment showed up to 7 ℃ warmer temperatures relative to satellite-derived SST. Such a warm anomaly might be caused by GDGT contributions from Euryarchaeota, which are known to dominate archaeal communities in the circumpolar deep water of the Antarctic Polar Front. The other reason might be that a linear calibration is not appropriate for this frontal region. Of the newly suggested SST proxies based on hydroxylated GDGTs (OH-GDGTs), only those with OH-GDGT−0 and Crenarchaeol or the ring index (RI) of OH-GDGTs yield realistic temperature estimates in our study regions, suggesting that OH-GDGTs could be applied as a potential temperature proxy in high latitude oceans.



Keywords: sinking particles, GDGT, TEX$_{86}^{L}$, Fram Strait, Antarctic Polar Front

# 1 Introduction

The knowledge that Thaumarchaeota, one phylum of Archaea, regulate the composition of their membrane lipids according to the surrounding water temperatures enabled the development of the paleothermometer TEX$_{86}$ (Kim et al., 2010; Schouten et al., 2002). TEX$_{86}$ is calculated based on the relative abundance of GDGTs containing 0−3 cyclopentane (GDGT−0~3) or 4 cyclopentane and one cyclohexane (Crenarchaeol) rings (Fig. 1).

The ubiquity of Thaumarchaeota, even in the polar oceans where the widely applied U$_{37}^{k'}$ proxy for sea surface temperature (SST) reconstructions is often problematic, supports the use of TEX$_{86}$ in high latitude regions (Bendle and Rosell-Melé, 2004; Ho et al., 2014). Nonetheless, the SST dependence of TEX$_{86}$ in warm regions is stronger than in colder regions where SSTs are below 5 °C (Kim et al., 2010). Thus, a logarithmic calibration, excluding the Crenarchaeol region-isomer, of TEX$_{86}^{L}$ was suggested for regions where maSSTs are below 15 °C (see Eq. 2; Kim et al., 2010). The authors speculated that the lack of correlation between the Crenarchaeol regio-isomer and SST at low temperatures might be caused by genetically different GDGT producers.

It has been questioned whether temperature is the only factor influencing the lipid composition (i.e., TEX$_{86}$) particularly in the regions where strong cold or warm biases were observed between TEX$_{86}$ reconstructed and measured maSSTs. The biases of TEX$_{86}$ calibrations have been attributed to enhanced production of GDGTs during seasons with favorable growth conditions for Thaumarchaeota (Pitcher et al., 2011; Wuchter et al., 2006b), a contribution of GDGTs from deep dwelling communities (Kim et al., 2015; Lee et al., 2008; Taylor et al., 2013), terrestrial input (Weijers et al., 2006), influences of different archaeal communities (Lincoln et al., 2014; Turich et al., 2007), and/or other environmental factors (Huguet et al., 2006; Mollenhauer et al., 2015; Park et al., 2018; Turich et al., 2007).

To circumvent these problems, especially in cold regions, a handful of studies have developed dedicated calibrations on regional scales (Meyer et al., 2016; Seki et al., 2014; Shevenell et al., 2011). For the temperature estimate in the Eocene Arctic Ocean, a modified version of TEX$_{86}$, the index termed TEX′$_{86}$ was proposed (Sluijs et al., 2006). It excludes GDGT−3 to eliminated contribution from terrestrial input, leading to significantly higher temperature estimates compared to the original TEX$_{86}$ calibration. The TEX′$_{86}$ values of 104 marine surface sediments showed a good linear correlation with maSSTs ($R^2$ = 0.93) (Sluijs et al., 2006). Following the addition of 7 additional TEX$_{86}$ values from surface sediments collected at the continental margin of the western Antarctic Peninsula, Shevenell et al., (2011) modified the Kim et al., (2008)'s TEX$_{86}$ calibration for this region (TEX$_{86}$ = 0.0125 × Temp + 0.3038, $R^2$ = 0.82) and estimated water temperatures for the Holocene.

Ho et al., (2014) evaluated TEX$_{86}$ and the different calibrations in an extended set of surface sediments from the polar regions of both hemispheres and found that the global TEX$_{86}$ calibration is suitable for subpolar regions (e.g., the Pacific





sector of the Southern Ocean and the Subarctic Front in the North Pacific). Additionally, the $TEX_{86}^L$ yields reasonable temperature estimates if regional calibrations are developed for high latitude regions. Using a regression of water temperatures (at 20 m depth in August) versus $TEX_{86}^L$ values, Seki et al., (2014) obtained plausible paleotemperature estimates for the Sea of Okhotsk and the subpolar North Pacific region. Meyer et al., (2016) confirmed later that this regional

$TEX_{86}^L$ calibration can also be applied to the subarctic northwest Pacific and the Western Bering Sea.

More recently, hydroxylated GDGTs (OH-GDGTs), which contain an additional hydroxyl group in one of the biphytanyl moieties of GDGT−0, −1, and −2, have been recognized in marine sediments (Liu et al., 2012; Fig. 1). Based on the finding that the abundance of OH-GDGTs relative to the total isoprenoid GDGTs increase towards cold regions (Huguet et al., 2013), OH-GDGT based SST calibrations have been proposed for high latitude oceans (Fietz et al., 2013; Huguet et al., 2013).

Beside the biological, phylogenetical, and statistical studies of the temperature proxy, sinking particles collected in specific regions can provide new perspectives on the distribution of GDGTs and $TEX_{86}$ based temperature reconstructions. The study using time-series sediment traps can give insights on the seasonal variability of lipid flux and temperature signals, and allows comparing the lipid signal to those in underlying sediments. Only one $TEX_{86}$ study using a time-series sediment trap in high latitude regions has been conducted near Iceland (Rodrigo-Gámiz et al., 2015), regardless of the interest in polar regions,

where temperature reconstructions with lipid proxies are troublesome. Here, we examine the seasonal GDGT production and $TEX_{86}^L$-derived temperatures in comparison to measured SSTs in order to better understand the drivers influencing $TEX_{86}^L$ values in sinking particles collected in two high latitude oceans (eastern Fram Strait, 79° N and Antarctic Polar Front of the Atlantic sector, 50° S). The temperatures derived from OH-GDGT proxies were also calculated to evaluate the applicability of these novel proxies for high latitude regions.

## 2 Study area

### 2.1 Fram Strait

The Fram Strait is located west of Spitsbergen, where the heat exchange occurs between the Arctic Ocean and the North Atlantic (Fig. 2). Relatively warm and nutrient-rich Atlantic water (AW) is transported by the West Spitsbergen Current (WSC) to the central Arctic Ocean, while colder and less saline Arctic water is transported along the East Greenland Current

(EGC) to the Nordic Seas (Fig. 2). Part of the WSC is separated from the northward flow and recirculated within the region (Manley, 1995; Soltwedel et al., 2016). Eddies, mixing, and recirculating Atlantic water contribute to the hydrographic complexity observed in the Fram Strait (Walczowski, 2013). The volume transport of the WSC and AW within the WSC displays strong seasonality with maxima in March and minima in July for the WSC, and maxima in late autumn and winter and minima in June for the AW (Beszczynska-Möller et al., 2012b). The occurrence of sea ice and its variability has an

effect on the particle fluxes as well as on benthic ecosystems (Bauerfeind et al., 2009; Hebbeln and Wefer, 1991; Soltwedel et al., 2016). Sinking particles in the Fram Strait, including organic and terrigenous material, are well known to originate not



only from the photic zone in the upper water column, but also from the Svalbard archipelago and Siberian shelf, from where they are transported by sea ice (Hebbeln, 2000; Lalande et al., 2016). Phytoplankton blooms enhance the vertical flux of biogenic components (organic carbon, carbonate, and opal) in spring-summer, but the downward flux of particles is likely affected by the environmental conditions (e.g. hydrographic changes, sea ice extent, and atmospheric low-pressure) on an

annual time scale (Wassmann et al., 2006). Based on oceanographic measurements from mooring arrays at multiple depths across the Fram Strait (from 7° W to 9° E, 78.3° N), Beszczynska-Möller et al., (2012) found two warm anomalies in the Atlantic water through the Fram Strait in 1999−2000 and 2005−2007. During the second warm anomaly, community structures of phytoplankton and zooplankton were affected by the decreased sea ice in this region (Lalande et al., 2013; Nöthig et al., 2015) and had a profound influence on biogenic sediment fluxes.

**2.2   Antarctic Polar Front**

The Antarctic Polar Front (APF) region is one of the major frontal zones within the westerly wind driven Antarctic Circumpolar Current (ACC) (Fig. 2). Spatial and temporal variability of the APF is strongly affected by seafloor topography. The APF has an average width of 43 km (1987−1993), which moves southward during the austral summer and reaches its northernmost position during the austral winter (Moore et al., 1999). The ACC is well known for its meandering jet flow and

eddy formation (Moore et al., 1999; Orsi et al., 1995), which likely stimulate phytoplankton blooms and primary production in this region by supplying growths limiting nutrients, and/or bringing deeper dwelling phytoplankton into the photic zone (Abbott et al., 2001; Moore and Abbott, 2000, 2002). Thus, the hydrographic structure is an important factor controlling the distribution of phytoplankton and small zooplankton in this region (Read et al., 2002). The highest chlorophyll-a levels of up to 3.5 mg m$^{-3}$ were recorded in the APF during the early austral summer 1990−1991, and it co-occurred with elevated

silicate levels as well as water column stability (Laubscher et al., 1993). The current speed close to the PF3 trap was recorded to be up to 8 cm s$^{-1}$ (Walter et al., 2001). Read et al., (2002) reported the vertical profile of currents measured between the surface and 400 m depth in the polar frontal regions in December 1995. The currents recorded at the surface and subsurface of the Polar Front were much stronger (30−50 cm s$^{-1}$) than the ones measured at 700 m depth (1−8 cm s$^{-1}$) by Walter et al., (2001).

**3   Material and Methods**

**3.1   Sediment Traps**

A time-series sediment trap system was moored to collect sinking particles at the eastern Fram Strait (FEVI16) between July 2007 and July 2008 at 1296 m water depth (Table 1, Fig. 2). Two traps (PF3) were deployed in the permanent ice free-area of the APF in the Atlantic sector at 614 m and 3196 m water depth from November 1989 to December 1990 (Table 1, Fig. 2).

The cone-shaped funnel of the Kiel trap systems had a 0.5 m$^2$ collection area, and the collection periods of each individual sample cup were programmed depending on the expected time of ice cover and/or the seasonality of the production. The





sampling cups were filled with filtered sea water enriched in sodium chloride (NaCl) to achieve a salinity of 40 psu, and they were poisoned with mercury chloride (HgCl$_2$; 0.14 % final solution) for sample preservation. For FEVI16 samples, zooplankton 'large swimmers' were removed using forceps before splitting each sampling cup into smaller sub-samples (Lalande et al., 2016). For PF3 samples, large swimmers were removed using forceps and a sieve (mesh size: 1 mm mesh) in

the laboratory (Fischer et al., 2002). Afterward, the samples were split for different purposes. A current meter (RCM 9/11) was attached to both trap moorings, and the data sets are available on PANGAEA (https://doi.org/10.1594/PANGAEA.845610; FEVI16) and in Walter et al., (2001; PF3)

## 3.2   Mass flux

Split FEVI16 samples were filtered onto GF/F filters (pre-combusted at 500 ℃ for 4 h) for organic carbon, organic nitrogen

(PON), and carbonate, and onto cellulose acetate filters for biogenic opal quantification. Organic carbon (POC) and organic nitrogen (PON) were measured with a CHN elemental analyzer after 0.1 N HCl treatment. Carbonate was calculated by subtracting the weight from total mass after 0.1 N HCl treatments and re-corrected according to the aragonite contents. The detailed methods used for biogenic silica can be found in Lalande et al., (2016). Split PF3 samples were freeze-dried for further processing. Decalcified samples (6 N HCl) were analysed for POC using a CHN elemental analyzer. Total nitrogen

(TN) was also determined. Carbonate was calculated by subtracting POC from total carbon (TC), which was directly measured without decalcification (carbonate = 8.33 × (TC - POC)). The methodology for the quantification of biogenic silica is described in Fischer et al., (2002).

## 3.3   GDGT analyses

For GDGT analysis, total lipids were extracted with a solvent mixture (9:1 v/v dichloromethane (DCM): methanol (MeOH))

using an ultrasonic bath for 10 min and a centrifuge for 5 min, after which the supernatant was decanted. This process was repeated three times and the supernatants were combined. Before the extraction process, a known amount of C$_{46}$-GDGT (internal standard) was added to each sample for GDGT quantification.

Following saponification of total lipids with 1 mL of 0.1 M potassium hydroxide (KOH) in a mixture of MeOH: purified water 9:1 (v/v) at 80 ℃ for two hours, neutral lipids (NLs) were recovered with 1 mL hexane (3 times). NLs were separated

into F1 (apolar), F2 (ketone), and F3 (polar) polarity fractions eluted in 2 mL hexane, 4 mL DCM: hexane (2:1 v/v), and 4 mL DCM: MeOH (1:1 v/v), respectively, using deactivated silica-gel chromatography (mesh size: 70-230 µm).

The F3 polar fraction containing GDGTs was re-dissolved in 500 µL hexane: isopropanol 99:1 (v/v) and filtered through a polytetrafluoroethylene (PTFE) filter (pore-size: 0.45 µm) into a glass insert of a 2 mL vial according to Hopmans et al., (2000). The filtered polar fraction was diluted with hexane: isopropanol 99:1 (v/v) to a concentration of approximately 2

mg/mL before the instrumental analysis.

GDGTs were analyzed using high-performance liquid chromatography/atmospheric pressure chemical ionization-mass spectrometry (HPLC/APCI-MS) according to Chen et al., (2014). Molecular ions *m/z* 1302, 1300, 1298, 1296, 1292 for





isoprenoid GDGTs and 1050, 1036 and 1022 for branched GDGTs were determined and quantified in relation to the molecular ion $m/z$ 744 of the $C_{46}$-GDGT. The late eluting peaks of OH-GDGT$-0$, $-1$, and $-2$ with $m/z$ 1318, 1316, and 1314 were also determined in the $m/z$ 1300, 1298, and 1296 scan, as described by Fietz et al., (2013).

### 3.4  GDGT flux and indices

5    $TEX_{86}^L$ values were calculated according to Kim et al., (2010).

$$TEX_{86}^L = Log_{10}\frac{[GDGT-2]}{[GDGT-1]+[GDGT-2]+[GDGT-2]} \tag{1}$$

where the numbers represent the number of cyclopentane moieties in the isoprenoid GDGTs.

$TEX_{86}^L$ values were converted into temperatures using the following equation (Kim et al., 2010):

$$SST(°C) = 67.5 \times TEX_{86}^L + 46.9 \ \left(R^2 = 0.86, n = 396\right) \tag{2}$$

10   GDGT flux represents the sum of fluxes of individual GDGTs, which are used for calculating $TEX_{86}^L$.

$$GDGT \ flux = [GDGT-1] + [GDGT-2] + [GDGT-3] \tag{3}$$

OH-GDGT based indices and estimated temperatures were calculated using the following equations (Eq.4 by Fietz et al., 2013; Eq. 5 and 6 by Lü et al., 2015):

$$\frac{[OH-GDGT-0]}{[Cren]} = 0.25 - 0.025 \times SST(°C) \ (R^2 = 0.58, n = 13) \tag{4}$$

15   $$RI-OH' = \frac{[OH-GDGT-1] + 2 \times [OH-GDGT-2]}{[OH-GDGT-0] + [OH-GDGT-1] + [OH-GDGT-2]} \tag{5}$$

$$RI-OH' = 0.0382 \times SST(°C) + 0.01 \ (R^2 = 0.75, n = 107) \tag{6}$$

BIT, an index of relative contribution of terrestrial versus marine input, was calculated according to Hopmans et al., (2004).

$$BIT = \frac{[I+II+III]}{[I+II+III+Cren]} \tag{7}$$



The Roman numerals I, II, and III refer to the branched GDGTs with 4, 5, and 6 methyl moieties, and Cren represents Crenarchaeol containing 4 cyclopentane moieties and 1 cyclohexane ring (Fig. 1). A terrestrial effect on $TEX_{86}$ can be significant when the BIT value is >0.3 (Weijers et al., 2006).

MI, the methane index indicating the relative contribution of GDGTs derived from methanotrophic Archaea to those from

planktonic Thaumarchaeota was calculated as follows (Zhang et al., 2011):

$$MI = \frac{[GDGT-1]+[GDGT-2]+[GDGT-3]}{[GDGT-1]+[GDGT-2]+[GDGT-3]+[Cren]+[Cren']} \tag{8}$$

Cren´ represents the regio-isomer of Crenarchaeol. Contributions from methanotrophic Archaea are considered to be significant when MI>3.

%GDGT-0, an indicator of a methanogenic source of GDGTs with a %GDGT value >67 %, was calculated as follows (Inglis

et al., 2015):

$$\%GDGT-0 = \frac{[GDGT-0]}{[GDGT-0]+[Cren]} \times 100\% \tag{9}$$

RI, the ring index, is a tool for identifying a potential non-thermal influence on GDGT distributions. A sample´s ring index is defined as follows (Zhang et al., 2016):

$$RI_{sanple} = 0 \times [GDGT-0] + 1 \times [GDGT-1] + 2 \times [GDGT-2] + 3 \times [GDGT-3] + 4 \times [Cren] + 4 \times [Cren'] \tag{10}$$

$\left|\Delta RI\right|$ indicates the residual of a sample´s RI ($RI_{sample}$) from a calculated RI ($RI_{calculated}$) based on the global $TEX_{86}$−RI regression. $RI_{calculated}$ and $\left|\Delta RI\right|$ are defined as follows:

$$RI_{calculated} = -0.77(\pm0.38) \times TEX_{86} + 3.32(\pm0.34) \times (TEX_{86})^2 + 1.59(\pm0.10) \quad (R^2=0.87, n=531) \tag{11}$$

$$\left|\Delta RI\right| = RI_{calculated} - RI_{sample} \tag{12}$$

Potential non-thermal influences on $TEX_{86}$ can be recognized when $\left|\Delta RI\right|$ is >0.3 (Zhang et al., 2016).

**3.5   Environmental parameters**

$TEX_{86}^{L}$-derived temperatures were compared to satellite-derived SSTs obtained by Advanced Very High Resolution Radiometer (AVHRR), which has a spatial grid resolution of 0.25° and temporal resolution of 1 day (Reynolds et al., 2007). The satellite-derived SSTs were averaged over the collection period of each sample cup. The depth profiles of mean annual water temperature and nitrate concentration were obtained from the World Ocean Atlas 2013 (WOA13) representing

averaged values for the years 1955-2012 (Garcia et al., 2013; Locarnini et al., 2013).



## 4    Results

### 4.1    Eastern Fram Strait: FEVI16

#### 4.1.1 Mass and GDGT fluxes

The mass flux data of the FEVI16 trap have been published previously by Lalande et al., (2016). In summary, the mass flux

(opal + POC + PON + lithogenic fluxes) measured at 1296 m water depth varied in between 18.5−652.9 mg m$^{-2}$ d$^{-1}$, with minima in the winter season (November 2007−February 2008; Fig. 3a). Lithogenic flux is presented separately in Fig. 3a due to its significantly higher contribution to total mass flux. The composition of sinking particles changed between seasons. Lithogenic material flux showed sudden pulses in spring (March and April−May 2008). POC was predominant in mid-July 2007. Carbonate flux was elevated in March, early-May and mid-June 2008. Opal was dominant in early-September 2007

and mid-May to June 2008.

GDGT fluxes varied between 7.2−85.9 ng m$^{-2}$ d$^{-1}$ (Fig. 3b). Peaks of GDGT flux occurred in August−September 2007 and March, May−June 2008, while minima were observed in the winter season (November 2007−February 2008). The GDGT flux was strongly correlated with opal flux (R$^2$ = 0.82, p<0.0001) when two GDGT maxima in late-September 2007 and March 2008 were excluded, and with carbonate flux (R$^2$ = 0.86, p<0.0001) except for the last two months in May-June 2008

(Fig. 3a and b).

#### 4.1.2 TEX$_{86}^{L}$ thermometry

TEX$_{86}^{L}$ values varied between -0.70−-0.64 and their TEX$_{86}^{L}$-derived temperatures (Kim et al (2010) calibration) ranged between 0.6−3.7 °C (Fig. 3b and f). The flux-weighted mean TEX$_{86}^{L}$ temperature during the trap deployment period was 2.8 °C. Two surface sediment TEX$_{86}^{L}$ values (-0.66 at PS68-251/2 (79.1° N, 4.6° E) and -0.65 at PS68-271/2 (79.3° N, 4.3° E))

are available near the FEVI16 trap (Ho et al., 2014). Most index values of BIT, MI, %GDGT-0, and │ΔRI │for samples from FEVI16 were below the critical values of each index (>0.3 for BIT and MI, >67 % for %GDGT-0, >0.3 for │ΔRI│). One sample with a │ΔRI │value >0.3 was found, indicating a non-thermal effect on TEX$_{86}$ (Table A1).

#### 4.1.3 Fractional abundance of OH-GDGTs

All OH-GDGTs were clearly present in FEVI16 samples. OH-GDGT-0 was predominant in the OH-GDGT pool, accounting

for 87−95 % (Fig. 4a). The proportion of OH-GDGTs in the sum of iso- and OH-GDGTs ranged between 7 and 11 % during the trap deployment period (Fig. 4a). Estimated temperatures calculated using the calibrations as described in Eq. (4) and (6) (Fietz et al., 2013 and Lü et al., 2015) ranged between -0.3−2.5 °C and between -0.9−1.4 °C, respectively. The last two data points for the trap time series were unusually low when using Eq. (4) (Fig. 4b).





## 4.2 Antarctic Polar Front: PF3

### 4.2.1 Mass and GDGT fluxes

The mass flux data of the PF3 trap have been published previously by Fischer et al., (2002). The mass flux at the shallow trap ranged between 19.7−592.1 mg m$^{-2}$ d$^{-1}$ with a distinct seasonal variability (Fig. 5a). Flux maxima occurred in

November 1989 to early-March 1990 and mid-October to November 1990, while it stayed relatively low from mid-March to mid-October 1990. At the deep trap, the mass flux varied between 0−516.6 mg m$^{-2}$ d$^{-1}$ (Fig. 5d). The mass flux peaked in mid-December 1989−January and March 1990 but stayed low for the remaining sampling period except for mid-May 1990. There was no clear correlation between organic material and GDGT fluxes at neither depth. The fluxes in GDGTs ranged from 10.5 to 73.9 ng m$^{-2}$ d$^{-1}$ at 614 m and from 0.5 to 153.9 ng m$^{-2}$ d$^{-1}$ at 3196 m depth (Fig. 5b and e). GDGT fluxes were

mostly enhanced during the low mass flux season at the shallow trap (Fig. 5b). GDGT flux maxima occurred in November and December 1989, late-May and early-November 1990 at the deep trap (Fig. 5e). GDGT analysis was not possible between late-July and October 1990 in the deep trap, since very little material was collected during this time period.

### 4.2.2 TEX$_{86}^{L}$ thermometry

TEX$_{86}^{L}$ values varied between -0.58−-0.67, and -0.55−-0.59 at the shallow and deep trap, respectively (Fig. 5b and e). The

TEX$_{86}^{L}$-derived temperatures ranged between 1.3−7.8 ℃ for the shallow trap, and 7.4−9.8 ℃ for the deep trap (Fig. 5c and f). The flux-weighted mean TEX$_{86}^{L}$ temperatures were 4.6 ℃ and 8.5 ℃ at the shallow and deep traps, respectively. In some samples, GDGT-3 was below the detection limit. In these cases, TEX$_{86}^{L}$ values and estimated temperatures were not included on the figures. but GDGT flux was calculated. A surface sediment TEX$_{86}^{L}$ value of -0.56 (CHN-115-4-34; 51.00° S, 5.33° E) in the vicinity of the PF3 trap site (50.13° S, 5.83° E) can be found in the TEX$_{86}$ global calibration dataset (Kim et al., 2010),

and its TEX$_{86}^{L}$-derived temperature estimate is 9.1 ℃. Most samples did not exceed the index values indicating non-thermal impacts on TEX$_{86}$ (see Sec. 3.4). Two samples from the deep trap between early November and late December 1989, however, showed MI values between 0.3 and 0.5 (Table A1).

### 4.2.3 Fractional abundance of OH-GDGTs

All OH-GDGTs were clearly present in samples from both depths. OH-GDGT−0 was predominant in the OH-GDGT pool

(76−89 % for the shallow trap, 77−83 % for the deep trap; Fig. 6a and c). The proportion of OH-GDGTs in the sum of iso- and OH-GDGTs ranged from 4−7 % and 5−6 % at the shallow and deep trap, respectively, (Fig. 6a and c). Estimated temperatures calculated according to Eq. (4) and (6) ranged between 3.0−6.1 ℃ and 1.1−5.1 ℃ in the shallow trap (Fig. 6b). With the same calibrations, estimated temperatures varied between 3.6−5.8 ℃ and 2.8−5.2 ℃ in the deep trap (Fig. 6d).



## 5    Discussion

### 5.1    Eastern Fram Strait (79° N)

#### 5.1.1 GDGT flux and particle export

Organic matter (including GDGTs), formed in the upper ocean, is exported to the bathypelagic mainly as zooplankton fecal

pellets or marine snow aggregates (Fischer and Karakaş, 2009; Wuchter et al., 2006b). The velocities of the sinking aggregates vary in relation to their composition (Fischer and Karakaş, 2009; Iversen and Ploug, 2010) and physical characteristics. For instance, aggregates formed from coccolithophores are ballasted by carbonate and may sink faster than opal ballasted diatom aggregates (Iversen and Ploug, 2010). Still, fecal pellets formed from either coccolithophores or diatoms sink faster than fecal pellets formed from non-ballasted flagellates (Ploug et al., 2008a, 2008b). GDGTs have been

suggested to be preferentially incorporated in opal-dominated particles (Mollenhauer et al., 2015). This is in agreement with observed $TEX_{86}$-derived temperatures in sediment-trap samples off Cape Blanc being delayed relative to the SST signal, and this time delay was longer than for $U_{37}^{K'}$-derived temperatures (Mollenhauer et al., 2015).

GDGT fluxes co-varied with fluxes of biogenic- and non-biogenic components in the eastern Fram Strait (Fig. 3). GDGT fluxes were enhanced in summer 2007 and spring 2008 with two clear maxima in late September 2007 and March 2008 (Fig.

3b). Without those two distinct GDGT flux maxima, GDGT flux showed a good correlation with opal ($R^2 = 0.82$, $p<0.0001$) throughout the deployment period. GDGT fluxes co-varied also with carbonate from the beginning of the deployment until April 2008 ($R^2 = 0.8$, $p<0.0001$) (Fig. 3a and b). This agrees with the concept that GDGTs are transported by particles mainly containing opal and carbonate as previously shown in sediment trap studies (Chen et al., 2016; Huguet et al., 2007; Mollenhauer et al., 2015; Park et al., 2018; Yamamoto et al., 2012). This observation suggests that GDGTs are exported

together with diatoms (opal) and coccolithophores (carbonate). Supporting this, diatom and coccolithophore fluxes both, previously reported by Lalande et al., (2016), showed good correlations with GDGT fluxes ($R^2 = 0.64$, $p<0.001$ for diatoms and $R^2 = 0.68$, $p<0.0001$ for coccolithophores) when excluding two GDGT flux maxima in the correlations with opal fluxes. The flux of terrestrial biomarkers (campesterol+ß-sitosterol), previously analyzed by Lalande et al., (2016), also showed a positive correlation with GDGT flux ($R^2 = 0.64$, $p<0.001$). The sterols (campesterol and ß-sitosterol), used frequently to

assess the plant-derived organic matter input to aquatic systems (Moreau et al., 2002), reflect the input of terrestrial matter transported by sea-ice in the Fram Strait (Lalande et al., 2016). We thus assume that terrestrial material is aggregated into particles which carry GDGTs. However, the lithogenic flux, also representing terrestrial input, displayed a different trend with maxima only in mid-April−mid-May 2008, when the sea ice concentration abruptly increased (Fig. 3e). The lithogenic material seems to be mainly supplied by downslope export from the nearby Svalbard archipelago (Lalande et al., 2016), with

a significant input when the sea ice is present (Fig. 3e).

GDGTs might also be exported with fecal pellets after grazing of Archaea by zooplankton. Indeed, GDGTs have been found in decapod guts and intestines, and their abundance ratios appear to be unaltered during gut passage (Huguet et al., 2006). The fecal pellet carbon (FPC) flux from appendicularians contributed more to the total carbon flux than copepod FPC flux.



Both appendicularian and copepod FPC fluxes were higher during spring 2008 than during summer 2007 (Fig. 3d; Lalande et al., (2016)). The correlation of GDGT fluxes with appendicularian FPC fluxes ($R^2$ = 0.67, p<0.05 in mid-July to mid-October 2007) implies grazing of Archaea by the appendicularians. Appendicularians are well known to ingest micro-size particles like Archaea (<1 µm) efficiently through a fine mucus filter (Conley and Sutherland, 2017).

Overall, it is reasonable to assume that seasonal changes in the relative proportion of materials that GDGTs can be aggregated with, may result in variable export velocities of particles carrying GDGTs to deeper waters in the eastern Fram Strait.

### 5.1.2 Variability of $TEX_{86}^L$-derived temperature

$TEX_{86}^L$ temperatures varied (-0.2−3.7 °C) within a similar range as the satellite-derived SSTs (-0.1−3.4 °C) during the trap
deployment period in the eastern Fram Strait. Although it did not display a clear seasonality, the $TEX_{86}^L$ signal is most likely to reflect the surface water environments without non-thermal effects influencing the $TEX_{86}^L$ (Fig. 3e). First of all, the index values (e.g. BIT, MI, %GDGT-0, RI), which could indicate potential non-thermal factors influencing the distribution of GDGTs, suggest that $TEX_{86}^L$ temperatures should reflect the upper water column temperature (Table A1). The BIT index defined by Hopmans et al., (2004) as a tracer for the terrestrial organic matter input was very low (<0.02), showing that
negligible amounts of soil-derived GDGTs are entrained in sinking particles. MI (Zhang et al., 2011) and %GDGT-0 (Inglis et al., 2015) have been suggested as indicators for the impact of methanotrophic Archaea even though it was questioned if the latter index can be applied in marine settings (Inglis et al., 2015). Both index values were consistently lower than the respective critical values of 0.3 for MI and 67 % for %GDGT-0. │ΔRI│ was suggested by Zhang et al., (2016) as an indicator for the integrated non-thermal factors on $TEX_{86}$. Most │ΔRI│ values were lower than a value of 0.3, and only
three samples collected in late September 2007 and June 2008 satisfied the criteria of being influenced by non-thermal factors. Secondly, although significant portions of particles transported to the mid-depth and deep water in the eastern Fram Strait originated off Svalbard and the Barents Sea (Lalande et al., 2016), it is known that GDGTs are most likely to reflect the local conditions rather than being affected by lateral transport (Kim et al., 2009; Mollenhauer et al., 2008). Therefore, we conclude that GDGTs are mainly transported from upper waters and non-regional sources of particles do not play a
significant role for the $TEX_{86}^L$ estimate in sinking particles of the eastern Fram Strait.

$TEX_{86}^L$ temperatures varied strongly between mid-July−October 2007, with the minimum estimated temperature (0.6 °C) when the GDGT and terrestrial biomarker fluxes were highest (Fig. 3b, c, and f). This is the most productive period in the Fram Strait and also the period with the highest export fluxes (e.g. Lalande et al., 2016). The material collected by deep ocean sediment traps is a mixture of many types of aggregates with different composition and settling velocities. Therefore,
fluctuating $TEX_{86}^L$ temperatures might be due to an average of signals from previous (before mid-July, i.e. slowly sinking particles) and current seasons (mid-July to October, i.e. fast settling aggregates), which were exported via different





aggregation and sinking mechanisms with different horizontal displacements during particle descent. Due to the collection of aggregates sinking with different velocities, one sample cup in a sediment trap may collect GDGTs of different ages.

During the low flux period between November 2007−February 2008, $TEX_{86}^L$ temperatures were constantly lower than the satellite-derived SSTs (Fig. 3a, b, and f). These signals might be derived from GDGTs synthesized in mid-August to October,

which was a late-bloom period dominated by protists that do not produce biominerals, such as *Phaeocystis* sp., dinoflagellates, and nanoflagellates (Nöthig et al., 2015). Aggregates formed without biominerals have low sinking velocities (Iversen and Ploug, 2010; Ploug et al., 2008a, 2008b), which may explain why the satellite-derived SSTs from mid-August to October 2007 were similar to the $TEX_{86}^L$-derived temperatures in sinking particles collected during November 2007 and February 2008. In spring 2008, $TEX_{86}^L$ temperatures stayed relatively warm until mid-April and suddenly dropped in May

and June (Fig. 3f). The relatively warm $TEX_{86}^L$ temperatures in particles collected in March and April 2008 probably reflect the signal transported by GDGTs produced in November−December 2007. Like previous seasons, the lack of transporting materials for GDGTs delayed the $TEX_{86}^L$ signal. The sudden drop of $TEX_{86}^L$-derived temperatures in mid-May to June 2008 was matched with the season of the enhanced flux of biogenic materials, GDGTs, phytoplankton, zooplankton fecal pellets and terrestrial biomarkers at the trap when the sea ice concentration was enhanced (Fig. 3). Eddies around the ice-edge create

upwelling and downwelling, which breaks the stratified surface water and supplies nutrients to the upper water column, fostering phytoplankton blooms (Lalande et al., 2013). The sea ice-edge bloom probably enhanced the GDGT production followed by zooplankton grazing. Additionally, the terrestrial materials as ballast were more available, which were derived from sea-ice as it melted during spring 2008. This process caused the fast export of $TEX_{86}^L$ signal from the surface waters in mid-May to June 2008. Overall, the particles containing GDGTs captured at 1296 m depth in the eastern Fram Strait were

likely delivered by a combination of packaging mechanisms involving biological and non-biological transport materials with different delay times over the season. Moreover, most GDGTs are exported vertically from the upper waters and reflect a regional $TEX_{86}^L$ signal in the eastern Fram Strait, while significant particle supply from the south via lateral advection has been reported previously (Lalande et al., 2016). This result agrees with the finding that diatom and fecal pellet fluxes can be traced as the export of local production regardless of the lateral particle supply.

The average export velocity of particles containing GDGTs can be calculated by dividing the travel distance of particles (i.e., the depth of the sediment trap) by the temporal offset between $TEX_{86}^L$-derived temperature and the corresponding satellite-derived SST, greatly simplifying the complexity of sinking mechanisms and range of settling velocities throughout the season (Mollenhauer et al., 2015). When the $TEX_{86}^L$-derived temperatures are shifted by 82 days, we found the best fit to the satellite-derived SSTs (Fig. 7b). Therefore, we see that GDGTs synthesized in the eastern Fram Strait likely travel to the trap

(1296 m) approximately within 82 days. This translates into an average minimum export velocity of GDGT-containing particles of approximately 15 m d$^{-1}$ when assuming export from 30-80 m water depth (see below Sec. 5.1.3). We have assumed that the export velocity includes the time of GDGT synthesis and transport time to the trap. The estimated sinking velocity of 15 m d$^{-1}$ is within range (9−17 m d$^{-1}$) estimated at a similar water depth (~1300 m) in the filamentous upwelling



zone off northwest Africa (Mollenhauer et al., 2015) and only slightly lower than settling velocities found previously for aggregates from the Fram Strait (Wekerle et al., 2018). Based on the seasonal succession of peaks in appendicularian fecal pellets in the same mooring system, their sinking velocity was estimated to be 5- to 11- fold higher (15−35 days to 2400 m depth; Lalande et al., 2016) compared to GDGTs. This is a realistic scenario because of the fast sinking velocity of a fecal

pellet, being rather big and dense, and generally faster average export velocities to deeper ocean depths (Fischer and Karakaş, 2009; Iversen et al., 2017). Moreover, the average export velocity of GDGTs includes the time it takes to be incorporated into aggregates. A more rapid export of GDGTs to depth resulting in smaller temporal offsets of $TEX_{86}^L$-derived temperature to water temperature changes occurred in mid-May−June 2008 when the FPC flux of appendicularians was enhanced (Fig. 3d). It has to be noted that the export velocity we calculated here may represent an averaged velocity of all the aggregates

collected in the trap, as the export velocity of GDGTs can vary depending on the type of sinking materials GDGTs are incorporated into.

  The flux-weighted mean $TEX_{86}^L$ temperature over the mooring period was 2.8 ℃. In the vicinity of the site FEVI16 (79.03° N, 4.35° E), two $TEX_{86}^L$ values from surface sediments are available in the dataset published by Ho et al., (2014). The estimated temperatures based on $TEX_{86}^L$ are 2.8 ℃ and 2.3 ℃ in PS68-251/2 (79.30° N, 4.30° E) and PS68-271/2 (79.10° N,

4.60° E), respectively. Considering the error of the $TEX_{86}^L$ calibration (Kim et al., 2010), $TEX_{86}^L$ estimates are almost identical between sinking particles and surface sediments. This suggests that GDGTs synthesized in the upper waters are propagated through the water column into the sediment without significant alteration even though the sinking process of GDGTs in the eastern Fram Strait seems quite complicated as discussed above. Iversen et al., (2010) and Jackson and Checkley Jr. (2011) also suggested that most biological activities resulting in aggregate alteration and degradation occur

around the base of the photic zone. Our observations agree with previous studies, as $TEX_{86}$ displays consistent values at multiple depths and/or in the underlying surface sediments in various environmental regimes such as in the Arabian sea (Wuchter et al., 2006b), in the northwestern Pacific (Yamamoto et al., 2012), off Cape Blanc (Mollenhauer et al., 2015), near Iceland (Rodrigo-Gámiz et al., 2015), off southern Java (Chen et al., 2016), and in the northern Gulf of Mexico (Richey and Tierney, 2016).

**5.1.3 Potential depth of $TEX_{86}^L$ signal origin**

To determine the water depth where the $TEX_{86}^L$ signal originated, the flux-weighted mean $TEX_{86}^L$ temperature (2.8 ℃) was compared to the depth profile of nutrient concentrations ($NH_4^+$, $NO_2^-$, and $NO_3^-$) measured in late June to early July 2010, 2011, and 2013, and water temperature extracted from WOA13 (79.125° N, 4.375° E) in the eastern Fram Strait (Fig. 8). Unfortunately, ammonia data are unavailable in 2007 and 2008. The mean estimated temperature (2.8 ℃) in sinking

particles corresponds to the surface water in July−September or of 30−80 m subsurface waters in April−June (Fig. 8). Major production of GDGTs in both seasons is a possible scenario because GDGT flux peaked in both seasons (Fig. 3b). However, the latter period (April−June) is the more plausible season dominantly supplying GDGTs to the sediment if we consider the



delay time of the GDGT signal (approximately 82 days, see Sec. 5.1.2), which illustrates the initial time of GDGT production (Fig. 3b). Furthermore, Thaumarchaeota, the primary synthesizers of GDGTs in the ocean, are aerobic ammonia oxidizers (Könneke et al., 2005) and maximum GDGT concentrations have been found near the $NO_2^-$ maximum, where maximum rates of ammonia oxidation and nitrification occur (Beman et al., 2008; Hurley et al., 2016). The depths of

consumption of ammonia and production of nitrite as well as nitrate accumulation as a result of the nitrification process in the subsurface can be deduced from the nutrient profile (Fig. 8). Assuming that maximum thaumarchaeotal abundance occurs at the depth of highest substrate availability, we, therefore, infer that Thaumarchaeota mainly record the subsurface water temperature (30−80 m) during the warm season when the spring bloom may occur (April−June) in the eastern Fram Strait. Similar observations were made in the Sea of Okhotsk and the north-west Pacific (Seki et al., 2014) and in the western

Bering Sea (Meyer et al., 2016), where the $TEX_{86}^L$-derived temperature is attributable to the regional season and water depth of GDGT production. The temperatures measured by a temperature sensor attached to the mooring array at approximately 68 m depth (Beszczynska-Möller et al., 2012a), which is approximately in the middle of the depth interval (30−80 m) corresponded to the $TEX_{86}^L$ temperature (Fig. 8), support our interpretation. The flux-weighted mean $TEX_{86}^L$ temperature is well within the range of measured temperatures at this depth and it reflects well the temperatures in March−June as deduced

from the nutrient depth profile (Fig. 7c).

## 5.2 Antarctic Polar Front (50° S)

### 5.2.1 GDGT flux

At the shallow trap (614 m water depth) of site PF3, there was no clear correlation between fluxes of organic matter and GDGTs, which displayed their respective maxima in different seasons (Fig. 5a and b). Peaks of mass flux occurred in

November 1989−early-March 1990 and mid-October−December 1990, while GDGT fluxes were highest in mid-May−June and late July−August 1990 (late austral autumn and winter) (Fig. 5a and b). Similar observations were made in the North Sea, where the abundance of Thaumarchaeota (previously known as marine group I Crenarchaeota) and GDGT concentrations were high in winter time, supported by the seasonality of Thaumarchaeotal 16S rRNA and amoA gene abundances in that region (Herfort et al., 2007; Pitcher et al., 2011; Wuchter et al., 2006a). An austral "winter bloom" of planktonic Archaea

was also found near the Antarctic Peninsula (Church et al., 2003; Murray et al., 1999; Tolar et al., 2016). Austral winter blooms of planktonic Archaea can be explained by higher ammonia availability in winter, while the chemoautotrophic Archaea are outcompeted by photoautotrophic phytoplankton in spring and summer when the light is not limiting (Könneke et al., 2005; Pitcher et al., 2011). Laubscher et al., (1993) found that ammonia was highly depleted at the chlorophyll-a maximum across the Antarctic Polar Front in early austral summer, which might create less favorable conditions for

ammonia-oxidizing planktonic Archaea. Additionally, Thaumarchaeota are known to be sensitive to photoinhibition (Horak et al., 2017). Therefore, the austral winter maxima in GDGT flux at the shallow trap of PF3 might be a consequence of the



higher production of planktonic Archaea in surface waters of the APF during the time when ammonia is more available and photoautotrophs cannot compete due to light limitation.

At the deep trap (3196 m water depth) of site PF3, GDGT flux peaked in November 1989 while mass flux was most pronounced in March 1990 (Fig. 5d and e). Due to the lack of sinking particles captured in the deep trap in June−November

1990, it is unclear if a potential "winter bloom" of GDGTs was also exported to deeper waters. The trapping efficiency of PF3 was found to be below 50 % at the deep trap using [230]Th as flux proxy (Walter et al., 2001). The negligible mass flux could thus be caused by low trapping efficiency. However, Fischer et al., (2002) observed a similar seasonal flux pattern in the following years at the same location of site PF3 including periods of almost no flux in July−December at a location further to the south in the marginal winter sea-ice zone. Thus, it is assumed that a low trapping efficiency or a failure of the

trap system did not account for the lack of particle samples at the deep trap of PF3.

### 5.2.2 Variability of $TEX_{86}^L$-derived temperature

$TEX_{86}^L$-derived temperatures at the shallow trap did not display a clear seasonal variability. Only 1/3 of the data points were similar to the SSTs, while the remaining samples showed warm or cold biased temperatures relative to the satellite-derived SSTs during the sampling period (Fig. 5c). All samples in the deep trap displayed warmer $TEX_{86}^L$ temperatures by up to 7 °C

relative to the satellite-derived SSTs (Fig. 5f). As discussed above, the mismatch between lipid proxy-based temperature estimates and satellite-derived temperature could be explained by the delay time of the proxy signal, due to the time needed between lipid synthesis and incorporation into sinking aggregates plus the sinking time (Mollenhauer et al., 2015; Müller and Fischer, 2003; Park et al., 2018). However, without a marked seasonality of the estimated temperature, it is difficult to determine a delay time of the $TEX_{86}$ signal. The other explanation for the absence of covariance between the $TEX_{86}$ signal

and the satellite-derived SST is that we observe a mixed signal of Thaumarchaeotal GDGTs derived from surface and deep ocean. The warm biases have a tendency to occur during periods of lower GDGT flux and thus may include higher proportions of material from different sources (early-December, mid-June−July, September−November), which may be more dominant in the deep trap (see discussion below).

Warm-biased $TEX_{86}^L$-derived temperatures have been attributed to the contribution of different archaeal communities or

GDGT input from terrestrial sources, which may alter the composition of pelagic GDGTs, leading to unusual $TEX_{86}^L$-derived temperature estimates (Hopmans et al., 2004; Inglis et al., 2015; Zhang et al., 2011, 2016). For example, several studies showed that anomalous $TEX_{86}$ based temperature estimates can be caused by isoprenoid GDGTs produced by Group II Euryarchaeota as significant contributors to the archaeal tetraether lipid pool and, thus, $TEX_{86}$ (Lincoln et al., 2014; Schouten et al., 2008, 2014; Turich et al., 2007), a suggestion that is controversial. A prevalence of marine Group II

Euryarchaeota has been reported in Circumpolar Deep Waters (Alonso-Sáez et al., 2011) and in deep waters of the Antarctic Polar Front (López-García et al., 2001; Martin-Cuadrado et al., 2008; Moreira et al., 2004; Murray et al., 1999). Therefore, we would assume that GDGTs produced by deep-dwelling Euryarchaeota might have caused the warm-biased $TEX_{86}^L$ signal





at depth in the region. Alternatively, advection of particles from warmer ocean regions could potentially lead to warm biases. However, it is known that the impact of lateral transport of GDGTs on the local signal is insignificant (Kim et al., 2009; Mollenhauer et al., 2008).

BIT, MI, %GDGT-0, and │ΔRI│ were examined to evaluate non-thermal factors on GDGT composition and $TEX_{86}$ values
at both PF3 traps. However, none of the values for BIT, %GDGT-0, and │ΔRI│ exceeded the defined critical values (Table A1). MI values were higher at the deep trap than at the shallow one. The first two samples at the deep trap reached values of 0.3−0.5 MI, suggesting that GDGTs are derived from a mixture of non-methanotrophic and methanotrophic communities (Zhang et al., 2011). However, these two samples alone cannot fully explain the episodic and continuous warm-biased $TEX_{86}^L$ estimates at the shallow and deep trap. Therefore, those indices cannot explain the warm anomaly of $TEX_{86}^L$ estimates
in the APF. Like in the Gulf of Mexico (Richey and Tierney, 2016), a longer time series of samples would be helpful to investigate inter-annual variability, paired with a direct assessment of the archaeal community in the region.

The flux-weighted mean $TEX_{86}^L$ temperatures averaged over the trap deployment period were 4.6 and 8.5 °C at the shallow and deep traps, respectively. The $TEX_{86}^L$ temperature in surface sediment (CHN 115-4-34; 51.00° S, 5.33° E) from the vicinity of the PF3 trap was 9.1 °C, which is 4.5 °C warmer than the $TEX_{86}^L$ estimate temperature in the shallow trap (4.6 °C)
but is similar to the one in the deep trap (8.5 °C). This situation is different from the Fram Strait and other time-series trap studies which showed that $TEX_{86}$ values in different depths are almost identical (Chen et al., 2016; Mollenhauer et al., 2015; Wuchter et al., 2006b). This implies that there is a clearly different origin of GDGTs in particles collected in the upper (<~ 600 m) and deeper ocean, yet the main explanation of the warm-biased $TEX_{86}^L$ is still speculative.

## 5.3   Re-evaluation of $TEX_{86}^L$ calibration in polar oceans

In the two previous sections, the $TEX_{86}^L$ calibration developed by Kim et al., (2010) for regions where maSSTs are below 15 °C was applied. This calibration is based on data from 396 surface sediments in the global ocean. Several studies have expanded the $TEX_{86}$ and $TEX_{86}^L$ surface sediment dataset. We revisited the latest global surface sediment dataset containing 1095 surface sediment measurements (Tierney and Tingley, 2015 and reference therein) to recalculate the $TEX_{86}^L$ calibration and to examine the agreement of our results with the new calibration. From the dataset, the data points for which $TEX_{86}^L$
values cannot be calculated or which were pointed out as problematic by Ho et al., (2014) (c.f. BIT>0.3 or ~ 1.0 and GDGT concentration below detection limit) were excluded to avoid potential biases. With approximately two times more data points for the $TEX_{86}^L$ calibration than the original one (Kim et al., 2010; $n$ = 396), the new calibration again shows a linear correlation with maSSTs even though the correlation coefficient is slightly lower (Fig. 9; $TEX_{86}^L$ = 0.013 × SST - 0.657; $R^2$ = 0.80, $n$ = 744) and the residual standard error (±5.0 °C) is higher than those of the Kim et al., (2010)'s calibration ($R^2$ = 0.86,
±4.0 °C). It should be noted that our new linear calibration has SST on the x-axis as the control variable and $TEX_{86}^L$ value on the y-axis as the dependent variable because the composition of GDGTs, membrane lipids of Thaumarchaeota, (i.e., $TEX_{86}$/



$TEX_{86}^{L}$ value) is potentially affected by water temperature. This different regression might explain parts of the discrepancy between our and the original calibrations of Kim et al. (2010). More scattered plots towards the cold temperatures appear in the new calibration as shown in previous studies (Ho et al., 2014; Kim et al., 2010) (Fig. 9).

It is obvious that the relationship of maSSTs to $TEX_{86}^{L}$ values in high latitude northern (>50° N) and southern (>50° S)
oceans is significantly different when both regions are plotted separately (Fig. 9). The data from the north (>50° N) are scattered both on the right- and left-hand side of the regression line, resulting in potential warm and cold anomalies, respectively (Fig. 9). This may reflect various sources of GDGTs in the high latitude northern oceans, which have direct geographical connections to Eurasia and North America. Warm biased $TEX_{86}^{L}$ estimates in these regions might be caused by input of soil-derived GDGTs, which are picked up by the sea ice in the Arctic marginal seas (c.f. Laptev sea, Kara Sea,
eastern Greenland), and released while the sea ice melts. Alternatively, $TEX_{86}^{L}$-derived temperatures might reflect warmer subsurface temperatures in the warm season as shown in the subarctic North Pacific region with the regional calibration by Seki et al., (2014) and Meyer et al., (2016). Cold-biased $TEX_{86}^{L}$ estimates are mainly found in the Barents Sea, North Sea, and the Norwegian Sea. In the North Sea, a significantly enhanced abundance of planktonic Archaeal cells and high GDGT concentration in the winter time could account for colder $TEX_{86}^{L}$-derived temperature compared to SST (Herfort et al., 2007;
Pitcher et al., 2011; Wuchter et al., 2006a). Hence, regional calibrations in the high latitude northern oceans seem to be a valid approach.

By contrast, most of the data from the south (>50° S) show higher $TEX_{86}^{L}$ values than predicted by the regression line ($n = 95$). This explains the observed warm anomalies of $TEX_{86}^{L}$-derived temperature when using the original linear regression calibration. Instead of the global linear calibration, a polynomial one seems to be a better option in these regions (Fig. 9).
When excluding one data point in the winter sea ice-covered Southern Ocean, the correlation coefficient of the polynomial correlation is encouraging ($R^2 = 0.63$).

The data points of the flux-weighted mean $TEX_{86}^{L}$ in the FEVI16 trap and $TEX_{86}^{L}$ in the underlying surface sediment against maSSTs are both closely located to the linear regression line. It illustrates the applicability of the linear $TEX_{86}^{L}$ calibration in the eastern Fram Strait (Fig. 9). At the site PF3, the $TEX_{86}^{L}$-derived temperatures based on the polynomial calibration yield -
0.1 ℃, 1.9 ℃, and 2.2 ℃ at the shallow and deep trap, and in the underlying surface sediment, respectively (Fig. 9), which are colder than the estimates based on the linear calibration. The polynomial $TEX_{86}^{L}$ temperatures at the deep trap and on the sediment are slightly colder than maSST (2.4 ℃, the data can be found in Tierney and Tingley, (2015)), but less biased compared to the reconstructed temperatures based on the linear $TEX_{86}^{L}$ calibration (see Sec. 5.2.2). To test this polynomial calibration in the southern ocean (>50 ℃S), a further assessment needs to be made with down-core sediments in similar
regions, where the original $TEX_{86}^{L}$ estimate displays a warm bias.



## 5.4 Applicability of OH-GDGT related calibrations for SST estimates in the high latitude Atlantic Ocean

OH-GDGTs were determined in sinking particles of the eastern Fram Strait and of the APF. The proportions of OH-GDGTs to total GDGTs (sum of OH- and iso-GDGTs) ranged between 7−14 % in the eastern Fram Strait and 4−7 % in the APF (Fig. 4a, 6a, and 6c). This is a relatively smaller portion than in the Nordic Seas (~ 16 %; Fietz et al., 2013).

Several OH-GDGT based calibrations for SST have been suggested for the global ocean and the Nordic Seas (Fietz et al., 2013; Huguet et al., 2013; Lü et al., 2015). We applied those calibrations suggested by Fietz et al., (2013) and Lü et al., (2015). Only two calibrations (Eq. 4 and 6) yield realistic temperature estimates, which varied within a similar range of satellite-derived SSTs or $TEX_{86}^{L}$-derived temperatures (Fig. 4b, 6b and 6d). The former calibration considers relative abundances of OH-GDGT−0 and Crenarchaeol (Eq. 4). The latter is based on the RI-OH-GDGTs (Eq. 5 and 6). In the

eastern Fram Strait, both OH-GDGTs based temperatures showed similar changes with an increasing trend until late-April 2008 and decreasing temperatures in May 2008 except for the last two samples (Fig. 4b). In the APF, both OH-GDGTs based temperatures were also close to the satellite-derived SSTs and the $TEX_{86}^{L}$-derived temperatures at the shallow trap (Fig. 6b). Warm biases with the $TEX_{86}^{L}$ calibration did not occur with those OH-GDGTs calibrations at the deep trap (Fig. 6d). We speculated in Sec. 5.2.2 that the warm-biased $TEX_{86}^{L}$ temperatures relative to SSTs may be caused by GDGTs synthesized by

Euryarchaeota dwelling in deep waters of the APF. The methanogenic Euryarchaeota are known to produce GDGT−1, −2 and −3 and may alter the $TEX_{86}$ signal derived from pelagic Thaumarchaeota (Weijers et al., 2011). This might explain why only calibrations (OH-GDGT/Cren and RI-OHʹ), which do not contain GDGTs potentially originated from Euryarchaeota, show a correspondence to regional SSTs. At this stage, however, this has to be considered speculative. Moreover, it is not clear yet if OH-GDGTs are exclusively produced by Thaumarchaeota since OH-GDGTs were also detected in a culture of

Euryarchaeota (*Methanothermococcus thermolithotrophicus*) (Liu et al., 2012). Therefore, further research is needed to clarify various aspects of distribution, production, and modification of OH-GDGTs in response to physicochemical changes in the global ocean. Nonetheless, OH-GDGTs appear to be a potential temperature proxy in our two high latitude regions.

## 6 Summary and Conclusions

Sinking particles collected using time-series sediment traps allowed us to determine the variability of the downward GDGT

export and the environmental influence on $TEX_{86}^{L}$ thermometry in northern and southern high latitude regions of the Atlantic Ocean. We observed fundamentally different patterns between the eastern Fram Strait and the Antarctic Polar Front.

In the eastern Fram Strait, the seasonally different composition of sinking materials resulted in different sinking velocities of GDGTs, and thus the temporal offsets between $TEX_{86}^{L}$-derived temperatures and SSTs may vary by season. Although increased flux of terrestrial matter transported by sea-ice in intermediate and deep waters is well known in the region

(Lalande et al., 2016), $TEX_{86}^{L}$ thermometry does not seem to be affected by lateral advection of particles. The flux-weighted mean $TEX_{86}^{L}$ temperature in sinking particles was similar to the one in the underlying surface sediment, indicating that the

$TEX_{86}^L$ signal did not experience substantial changes while sinking. The flux-weighted mean $TEX_{86}^L$ temperature corresponded to temperatures in water depths ranging between 30 and 80 m, where nutrient profiles suggest favorable conditions for Thaumarchaeota.

In the Antarctic Polar Front, changes in GDGT fluxes and $TEX_{86}^L$ temperatures are different between the two traps moored at

different depths. At the shallow trap, $TEX_{86}^L$ estimates do not co-vary with satellite-derived SSTs, and its flux-weighted mean temperature is 4.6 °C. The warm-biased $TEX_{86}^L$ at the deep trap and in the underlying surface sediment may be caused by GDGT contributions from Euryarchaeota, which is dominant in the deeper part of the water column of CDW at the Antarctic Polar Front. Alternatively, a systematic warm bias of the linear calibration in the high latitude southern ocean could explain the discrepancy between $TEX_{86}^L$ reconstructed and observed temperatures. The discrepancies can be reduced by using a non-

linear $TEX_{86}^L$ calibration developed for high latitude samples from the Southern Hemisphere.

Our findings offer insights into the potential factors governing $TEX_{86}^L$ thermometry such as nutrient availability or archaeal community composition in high latitude regions. In the high latitude North Atlantic, regional $TEX_{86}^L$ calibrations can be an additional option next to the global calibration, while in the high latitude Southern Ocean, the benefit of a polynomial $TEX_{86}^L$ calibration needs to be further tested. Our studies suggest that OH-GDGTs based calibrations are also worth to be further

investigated in the polar oceans. Accordingly, our study highlights that multiple approaches of global versus regional calibrations, polynomial relationships, or OH-related calibrations are beneficial to overcome the limitations of a single $TEX_{86}^L$ calibration in high latitude ocean regions.

**Data availability**

The data presented here are available on the PANGAEA database (https://doi.pangaea.de/10.1594/PANGAEA.897268).

**Author contributions**

EP and GM designed the study. EP measured the samples with a help of JH, analyzed the data, prepared figures and tables, and wrote the manuscript. All the authors provided feedback on the manuscript and GM reviewed the manuscript.

**Competing interests**

The authors declare that they have no conflict of interest.





**Acknowledgments**

We acknowledge the captain, crew and scientists who participated in the expeditions for collecting sediment trap samples used in this study. This project is funded through the DFG-Research Center/Cluster of Excellence "The Ocean in the Earth System" at the MARUM and supported by GLOMAR−Bremen International Graduate School for Marine Sciences,

5    University of Bremen. The trap samples were supplied by the MARUM Centre of Marine Environmental Research at the University of Bremen (PF3) and Alfred-Wegener Institute, Helmholtz Center for Polar and Marine Sciences (FEVI16). MHI received funding for the Helmholtz Young Investigator Group SeaPump "Seasonal and regional food web interactions with the biological pump" (VH-NG-1000), the Alfred Wegener Institute for Polar and Marine Research.



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





**Table 1.** Information on FEVI16 and PF3 trap.

| Name | Region | Location | | Water depth | Trap depth | Deployment period | | Sampling interval | Cruise reports |
|---|---|---|---|---|---|---|---|---|---|
| | | Latitude (° N) | Longitude (° E) | | | Start | End | | |
| | | | | (m) | (m) | (dd.mm.yyyy) | | (d) | |
| FEVI16 | Eastern Fram Strait | 79.02 | 4.35 | 2589 | 1296 | 23.07.2007 | 30.06.2008 | 10-31* | ARK-XXII/1c (Klages and Participants, 2007) ARK-XXIII/2 (Kattner and Participants, 2009) |
| PF3 | Antarctic Polar Front | -50.13 | 5.83 | 3785 | 614 3196 | 10.11.1989 | 23.12.1990 | 21, 42* | ANT-VIII/3 (Gersonde and Participants, 1990) ANT-IX/2 (Fahrbach and Cruise Participants, 1992) |

\* The exact sampling interval of each sample at FEVI16 and PF3 can be found on PANGAEA
(https://doi.pangaea.de/10.1594/PANGAEA.897268).





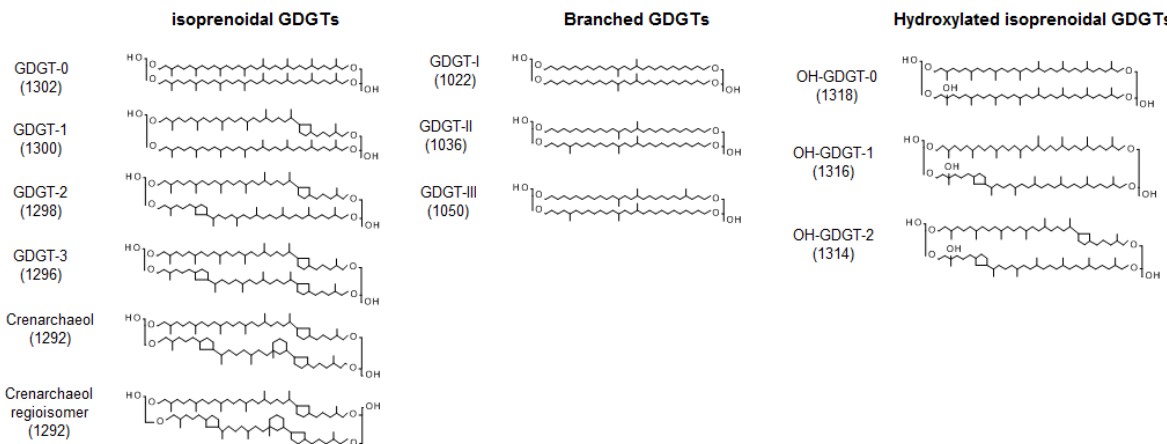

**Figure 1. Chemical structures and molecular ion m/z values of the isoprenoid glycerol dialkyl glycerol tetraethers (GDGTs), branched GDGTs, and hydroxylated GDGTs.**



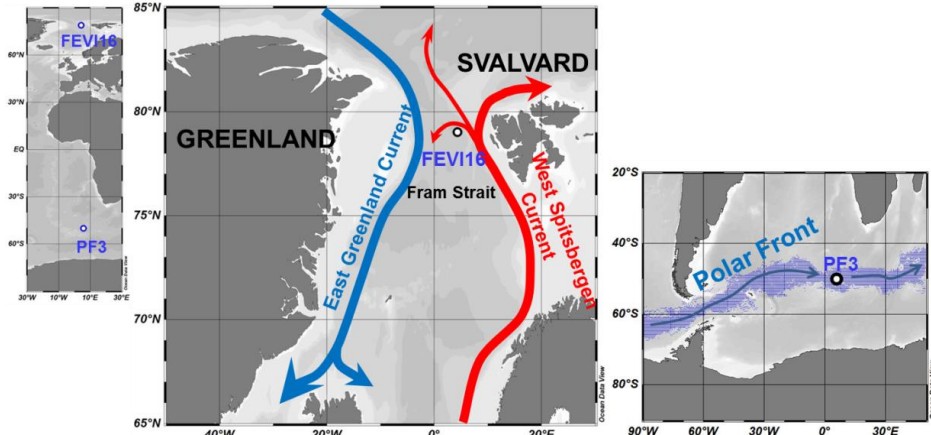

**Figure 2. Map of sediment trap locations in high northern latitude (FEVI16) and in Antarctic Polar Front (PF3). In Fram Strait, the northward red arrow represents the West Spitsbergen Current bringing warm and saline Atlantic Waters to the Arctic and the southward blue arrow displays the East Greenland Current transporting cold and fresh water to the Norwegian Sea. Around the Antarctic, the digitized blue shadow exhibits the location of the**
5 **Antarctic Polar Front for the year 2002-2014 at weekly resolution (Freeman and Lovenduski, 2016). The blue arrow indicates the clockwise Antarctic Circumpolar Current. Ocean Data View is used for mapping (Schlitzer, 2017; available at https://odv.awi.de).**





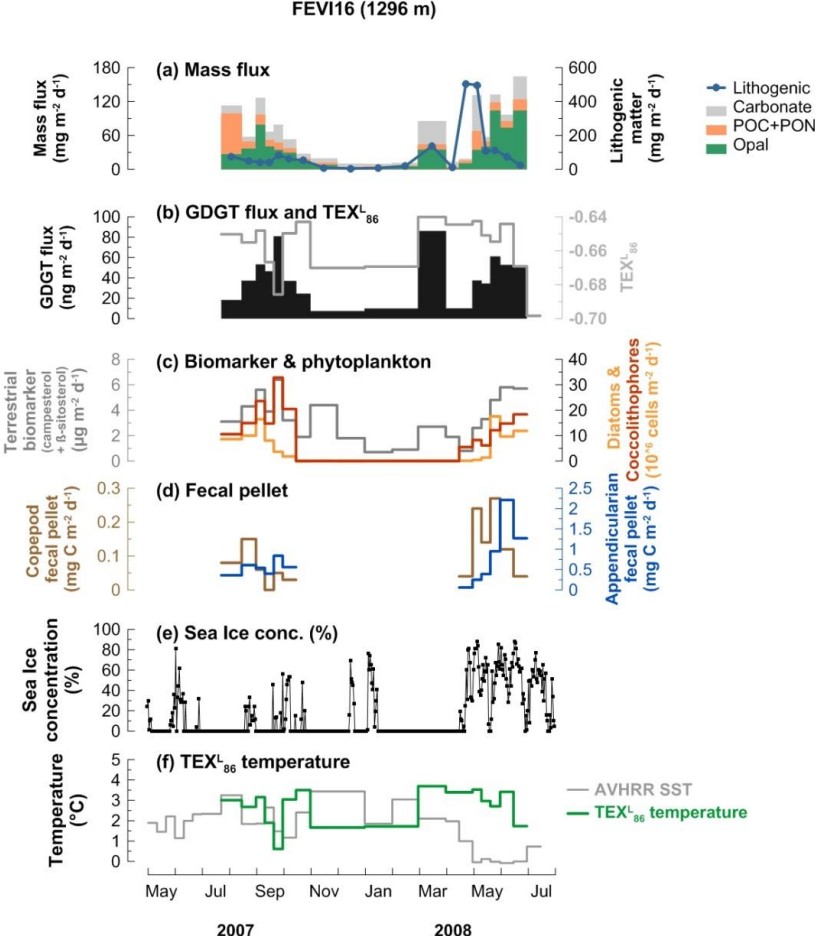

**Figure 3.** Changes in mass flux (a), GDGT flux (b), terrestrial biomarker flux and diatom and coccolithophore abundances (c), fecal pellet flux (d), Sea ice concentration (e), $TEX_{86}^{L}$-derived temperatures and satellite-derived SSTs obtained at 79.125° N, 4.375° E near the trap site (f) at 1296 m water depth in the eastern Fram Strait site, FEVI16 during the deployment period (July 2007−July 2008). (a), (c), (d), and (e) are previously reported by Lalande et al., (2016).



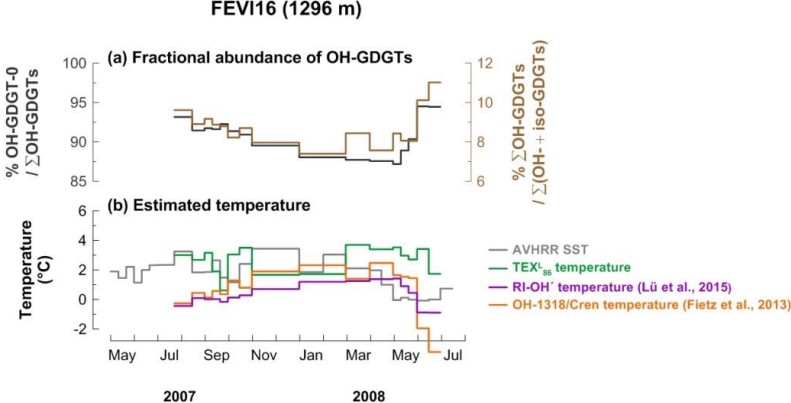

**Figure 4. Changes in fractional abundance of OH-GDGTs (a) and estimated temperatures based on OH-related calibrations (b) at site FEVI16. The purple and orange lines indicate the calculated temperatures using the calibrations by Fietz et al., (2013) (Eq. 4) and Lü et al., (2015) (Eq. 6), respectively. The green line indicates the TEX$_{86}^{L}$-derived temperatures.**





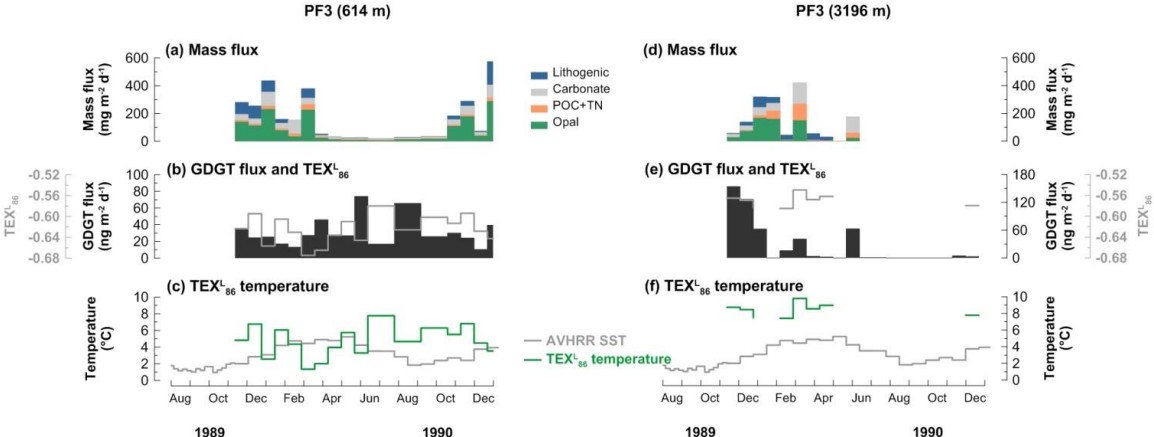

**Figure 5. Changes in mass flux (a, d), GDGT flux (b, e) and TEX$_{86}^{L}$-derived temperatures and satellite-derived SSTs obtained at 50.125° S, 5.875° E near the trap site (c, f) at the 614 m (left panels) and 3196 m (right panels) water depth in the Antarctic Polar Front site (PF3) during the deployment period (November 1989−December 1990).**





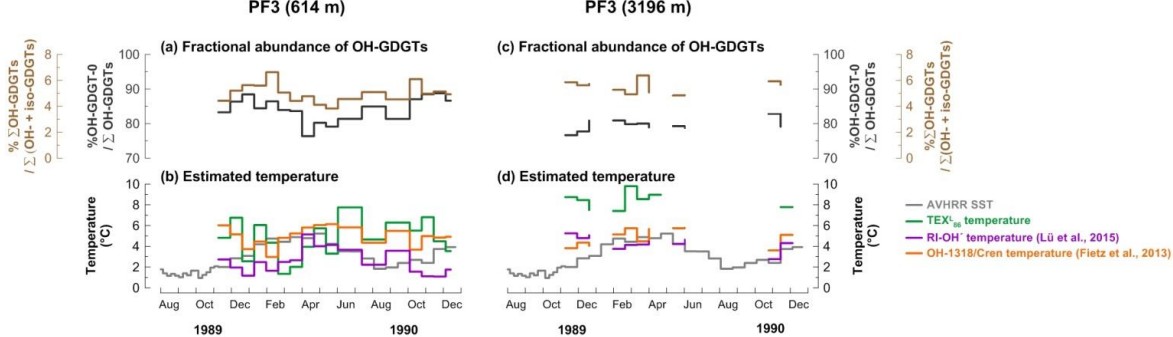

**Figure 6. Changes in fractional abundance of OH-GDGTs (a, c) and estimated temperatures based on OH-related calibrations (b, d) at site PF3. The purple and orange lines indicate the calculated temperatures using the calibrations by Fietz et al., (2013) (Eq. 4) and Lü et al., (2015) (Eq. 6), respectively. The green lines indicate the $TEX_{86}^L$-derived temperatures.**





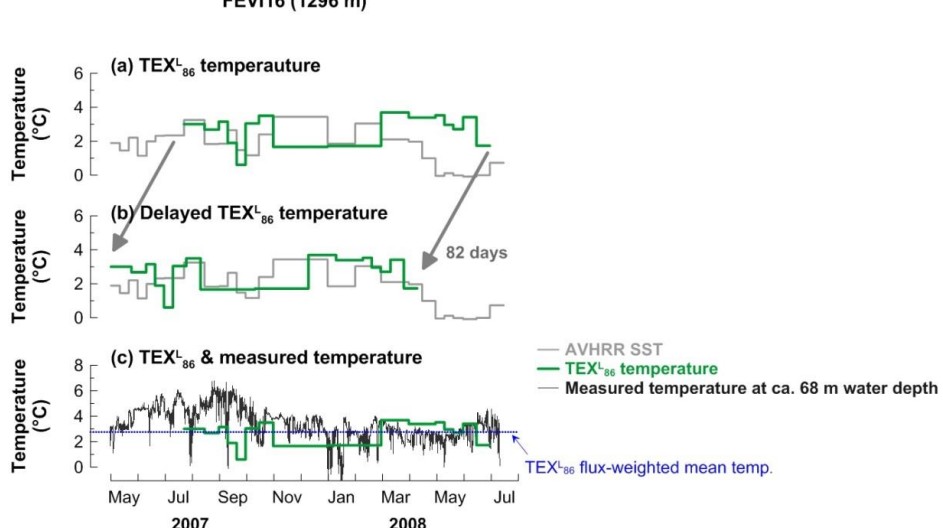

**Figure 7. Changes in $TEX^L_{86}$-derived temperatures and satellite-derived SSTs (a), time-delayed $TEX^L_{86}$ temperatures (b), sensor measured temperatures at approximately 68 m water depth at site FEVI16.**





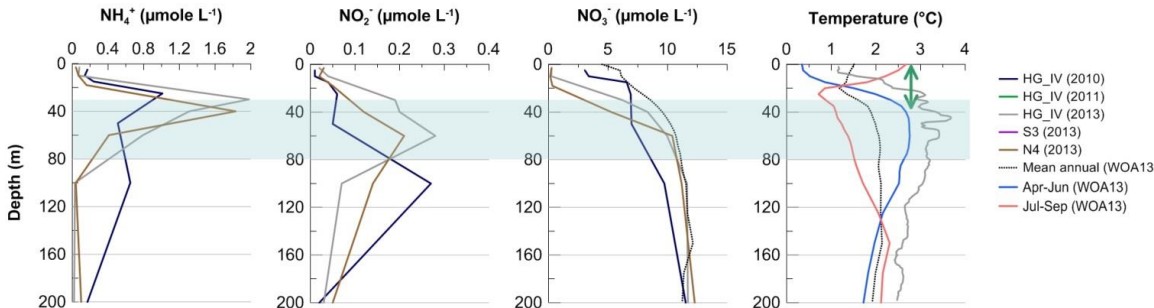

**Figure 8. Depth profiles of nutrient concentration ($NH_4^+$, $NO_2^-$, $NO_3^-$), and temperature in the Fram Strait. The navy, green, and gray lines represent the central station of the Long-Term Ecological Research (LTER) observatory HAUSGARTEN (HG-IV), close to FEVI16 trap location. The purple and brown lines represent the southern and northern stations of HG-IV. Black dotted lines for $NO_3^-$ and temperature, and blue and orange lines for temperature represent the data sets extracted from WOA13. Only two seasonal profiles (April to June and July to September) from WOA13 are available at the same location (79.125° N, 4.375° E) near the site FEVI16. Measured nutrient concentrations are reported on PANGAEA (Bauerfeind et al., 2014).**





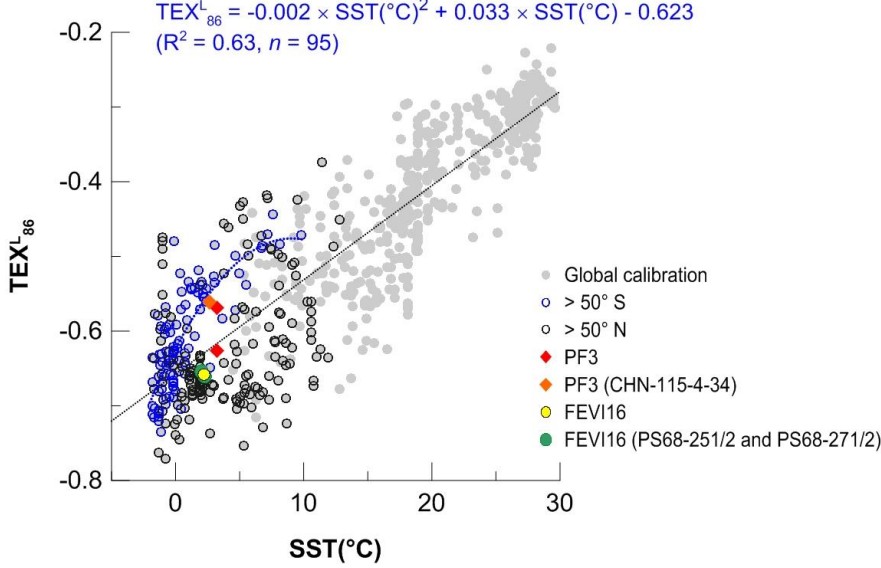

**Figure 9.** Correlation of TEX$_{86}^{L}$ values with mean annual SSTs in the global ocean (gray symbols; Tierney and Tingley, (2015) and reference therein). Blue and black round symbols represent the high latitude regions higher than 50° S and 50° N, respectively. In the southern regions (>50° S), TEX$_{86}^{L}$ values have a polynomial correlation with maSSTs (R2 = 0.63, n = 95). Red and orange diamonds represent the flux-weighted mean TEX$_{86}^{L}$ of the two
5  traps at the site PF3 and TEX$_{86}^{L}$ on the surface sediment (CHN-115-4-34), respectively. Yellow and green round symbols represent the flux-weighted mean TEX$_{86}^{L}$ at the site FEVI16 and TEX$_{86}^{L}$ on the surface sediments (PS68-251/2 and PS68-271/2), respectively.





**Appendix A.**

**Table A1. Index values for assessing the terrestrial and/or methanogen GDGT inputs at FEVI16 and PF3 sediment traps. The exact sampling interval of each sample number at FEVI16 and PF3 can be found in PANGAEA (https://doi.pangaea.de/10.1594/PANGAEA.897268)**

| Cup no. | FEVI16 (1296 m) | | | | PF3 (614 m) | | | | PF3 (3196 m) | | | |
|---|---|---|---|---|---|---|---|---|---|---|---|---|
| | BIT [a] | MI [b] | %GDGT-0 [c] | \|ΔRI\| [d] | BIT | MI | %GDGT-0 | \|ΔRI\| | BIT | MI | %GDGT-0 | \|ΔRI\| |
| 1 | 0.01 | 0.09 | 60 | 0.15 | 0.00 | 0.15 | 58 | 0.05 | 0.00 | 0.34 | 62 | 0.23 |
| 2 | 0.01 | 0.11 | 60 | 0.22 | 0.02 | 0.14 | 58 | 0.08 | 0.00 | 0.30 | 61 | 0.17 |
| 3 | 0.01 | 0.11 | 60 | 0.21 | 0.00 | 0.14 | 64 | 0.24 | 0.00 | 0.21 | 58 | 0.09 |
| 4 | 0.01 | 0.10 | 60 | 0.20 | 0.00 | 0.16 | 61 | 0.16 | - | - | - | - |
| 5 | 0.01 | 0.11 | 61 | 0.30 | 0.00 | 0.19 | 62 | 0.17 | 0.00 | 0.22 | 58 | 0.06 |
| 6 | 0.01 | 0.10 | 60 | 0.23 | 0.00 | 0.15 | 63 | 0.19 | 0.00 | 0.21 | 56 | 0.05 |
| 7 | 0.01 | 0.11 | 60 | 0.23 | 0.00 | 0.20 | 65 | 0.24 | 0.00 | 0.19 | 55 | 0.08 |
| 8 | 0.01 | 0.11 | 60 | 0.09 | 0.00 | 0.19 | 60 | 0.08 | 0.00 | 0.21 | 56 | 0.01 |
| 9 | | | | | 0.00 | 0.19 | 62 | 0.16 | | | | |
| 10 | 0.01 | 0.13 | 61 | 0.14 | 0.00 | 0.23 | 64 | 0.20 | - | - | - | - |
| 11 | | | | | 0.00 | 0.20 | 59 | 0.06 | - | - | - | - |
| 12 | 0.01 | 0.15 | 59 | 0.19 | 0.00 | 0.21 | 64 | 0.26 | - | - | - | - |
| 13 | 0.01 | 0.13 | 59 | 0.10 | 0.00 | 0.21 | 62 | 0.19 | - | - | - | - |
| 14 | | | | | 0.00 | 0.16 | 61 | 0.16 | - | - | - | - |
| 15 | 0.02 | 0.14 | 59 | 0.13 | 0.00 | 0.15 | 60 | 0.15 | - | - | - | - |
| 16 | 0.02 | 0.13 | 61 | 0.22 | 0.00 | 0.14 | 60 | 0.11 | 0.00 | 0.21 | 56 | 0.04 |
| 17 | 0.01 | 0.11 | 61 | 0.20 | 0.00 | 0.16 | 62 | 0.16 | - | - | - | - |
| 18 | 0.01 | 0.09 | 63 | 0.26 | | | | | | | | |
| 19 | 0.01 | 0.10 | 64 | 0.31 | | | | | | | | |
| 20 | - | - | - | - | | | | | | | | |
| | Total 20 samples | | | | Total 17 samples | | | | | | | |

[a]: BIT (Hopmans et al., 2004; Weijers et al., 2006) for terrestrial input when > 0.3
[b]: MI (Zhang et al., 2011) for methanotrophic archaeal input when > 0.5 and normal oceanic signal when < 0.3
[c]: %GDGT-0 (Blaga et al., 2009; Inglis et al., 2015) for methanogenic archaeal input when > 67 %
[d]: \|ΔRI\| (Zhang et al., 2016) for both methanotrophic archaeal and terrestrial input > 0.3