# Peer review of "Seasonality of archaeal lipid flux and GDGT-based thermometry in sinking particles of high latitude oceans: Fram Strait (79 $^{\circ}$ N) and Antarctic Polar Front (50 $^{\circ}$ S)"

_Biogeosciences, 2019_

## Referee Comment (RC1) · Anonymous Referee #1 · 13 Mar 2019

This is a very detailed assessment of the sinking dynamics and depth distribution of marine GDGTs and associated proxy indices at two high latitude locations. Particularly, I find it remarkable that OH-GDGT may be a promising alternative temperature proxy to GDGTs in these regions.

I have a couple of mostly minor comments/suggestion below. I am, however, a bit concerned about leaving out data points, without reasonable justification (see comments below).

[Figure]

P1-12 remove 'the'

P1-13 'proxies' and 'proxy index'

P1-15 '... where the original TEX86 proxy calibration shows a larger scatter.'

P1-21 remove 'during transport', it's redundant

P2-10 '... a logarithmic calibration of TEX86L, excluding the Crenarchaeol regio isomer, was suggested ...'

P4-20 'stratification' instead of 'stability'?

P5-05 'Afterwards'

P8-12-15 Why were these samples excluded? Is there reason to believe that something is wrong with the analyses? If not, the statistics should include all samples.

P9-25 check subscript

P10-6 '... vary depending on their composition... '

P10-10 '... preferentially incorporated into ...'

P10-20-22 Again, please provide reasoning for excluding samples from correlation analyses, or revise.

P12-21-22 What evidence do you base this statement on? Include explanation, or reference to figure.

P12-23-24 This argument is not quite clear to me. Which 'result' are you referring to?

P14-25-28 How exactly (over which nutrient) do you think Thaumarchaeota compete with phytoplankton? Does phytoplankton use ammonium as a N source?

P15-7-10 At which depth die Fischer et al. observe similar patterns? It is also not clear, which location are you referring to. Therefore, the conclusion you make is not clear either.

P16-30 This statement should be stronger (remove 'potentially'), because water T has an effect on GDGTs, and not the other way round.

P17-2 'Larger scatter towards colder temperatures ...' P17-4 '...relationship of maSSTs AND TEX86L values ...'

P18-7 'similar range as'

P18-13 'Warm biases AS with the ...'

Page: 19

P19-16-17 '... or OH-GDGT-based calibrations ... the limitations of a single global TEX86L calibration ... '

---

## Referee Comment (RC2) · Anonymous Referee #2 · 16 Mar 2019

The manuscript by Park et al. reports on seasonality of archaeal fluxes and GDGT-based thermometry in sinking particles based on two case studies in high latitudes. The study is based on material collected in sediment traps at different depths. This approach is complementary to the collection of surface sediments and offers the opportunity to study processes and mechanisms lying to the signal acquisition in the sediments. An interesting point is made on depths of production of OH-GDGTs. and the consequences on RI-OH thermometers. To conclude this manuscript address important issues in the paleo-proxy community and the new set of data presented is

interesting. I therefore recommend the publication of this manuscript with minor revisions detailed below: 1. A more throughout presentation of the errors associated with the temperature reconstruction based on the different indices and different calibrations should be discussed and provided. 2. Different processes of the production as well as the export of GDGTs are investigated in in two settings, even if the figures are already numerous, it would be interesting to provide the reader with a figure/sketch summarizing the mechanisms of production (seasonality/community or depth changes) and export (type of ballasts or timing) in the two settings.

---

## Author Comment (AC1) · 13 Apr 2019

Dear editor,

We thank the two anonymous referees for evaluating our manuscript. We are pleased to learn that both referees have found our work interesting. Their comments and suggestions helped us to improve the manuscript. Below are our replies to the referees' comments in blue. Page numbers mentioned here refer to the original manuscript published on Biogeosciences discussions.

**Anonymous Referee#1**

**Referee's comments (RC) -** This is a very detailed assessment of the sinking dynamics and depth distribution of marine GDGTs and associated proxy indices at two high latitude locations. Particularly, I find it remarkable that OH-GDGT may be a promising alternative temperature proxy to GDGTs in these regions. I have a couple of mostly minor comments/suggestion below. I am, however, a bit concerned about leaving out data points, without reasonable justification (see comments below: please see 8. Referee's comments (RC) and Author's responses (AR)).

**Specific comments**

1. **Referee's comments (RC)** - P1-12 remove 'the'
   **Author's responses (AR)** - Corrected

2. **RC** - P1-13 'proxies' and 'proxy index'
   **AR -** Corrected

3. **RC** - P1-15 '... where the original TEX86 proxy calibration shows a larger scatter.'
   **AR** - Corrected

4. **RC** - P1-21 remove 'during transport', it's redundant
   **AR** - Removed from the sentence

5. **RC** - P2-10 '... a logarithmic calibration of TEX86L, excluding the Crenarchaeol regio isomer, was suggested ...'
   **AR** - Corrected

6. **RC** - P4-20 'stratification' instead of 'stability'?
   **AR** - Corrected. Yes, 'stratification' is the proper term in this context rather than 'stability'.

7. **RC** - P5-05 'Afterwards'
   **AR** - Corrected

8. **RC** - P8-12-15 Why were these samples excluded? Is there reason to believe that something is wrong with the analyses? If not, the statistics should include all samples.
   **AR** - When all samples are considered, the relationship between GDGT and opal fluxes does not seem well correlated ($R^2 = 0.44$). However, by visual inspection it is evident that the changes in GDGT fluxes have a similar trend with the ones in opal flux. Moreover, the episodic GDGT pulses we excluded for the correlation occurred when carbonate (late September 2007 and March 2008), coccolithophore (late September 2007), and terrestrial biomarker (late September 2007) fluxes were enhanced. Therefore, we believe that the episodic GDGT pulses occurring at these times might have been a consequence of enhanced GDGT flux exported by those transporting materials (carbonate, coccolithophore, terrestrial matter) rather than by opal, simply because during these times a very high particle flux prevailed, enhancing the export of every material from the surface waters.
   Author's changes in manuscript - we added the sentences in the discussion section in P10 line16-18 "Those two episodic GDGT pulses occurred when carbonate, coccolithophore, and terrestrial biomarker fluxes were enhanced, potentially resulting in the enhanced GDGT flux."

9. RC - P9-25 check subscript
   **AR** - Corrected

10. **RC** - P10-6 '... vary depending on their composition... '

**AR** - Corrected

11. **RC** - P10-10 '... preferentially incorporated into ...'
    **AR** - Corrected

12. **RC** - P10-20-22 Again, please provide reasoning for excluding samples from correlation analyses, or revise.
    **AR** - Please see 8. AS

13. **RC** - P12-21-22 What evidence do you base this statement on? Include explanation, or reference to figure.
    **AR** - The reflection of SSTs based on the $\text{TEX}_{86}^{L}$ calibration shows evidence that GDGTs are mainly derived from surface waters. Figure 3f is referred to.
    Author's changes in manuscript – We refer to Figure 3f.

14. **RC** - P12-23-24 This argument is not quite clear to me. Which 'result' are you referring to?
    **AR** - To make the sentence clear, we rephrased it and inserted the reference. 'The result' means that GDGTs transported by diatom and fecal pellet reflect the SSTs based on the $\text{TEX}_{86}^{L}$ proxy and there is a good correlation between GDGTs and opal as well as Appendicularian fecal pellet fluxes.
    Author's changes in manuscript - 'The reflection of SSTs based on the $\text{TEX}_{86}^{L}$ calibration and correlations of GDGTs with opal and Appendicularian fecal pellet fluxes agree with the finding…….(Lalande et al., 2016).'

15. **RC** - P14-25-28 How exactly (over which nutrient) do you think Thaumarchaeota compete with phytoplankton? Does phytoplankton use ammonium as a N source?
    **AR** - Thaumarchaeota as ammonia-oxidizer compete with phytoplankton for ammonia, which use ammonia as a N source. To make the sentence clear, we rephrased it.
    Author's changes in manuscript – 'Austral winter blooms of planktonic Archaea…..' is changed to 'Photoautotrophic phytoplankton, which use ammonium as a N source, would outcompete Archaea for ammonia in spring and summer time. In contrast, in winter time when phytoplankton's productivity is limited due to the lack of light, ammonia availability for Archaea is higher (Pitcher et al., 2011; Wuchter et al., 2006). This explains the winter bloom of ammonia-oxidizing Archaea.'

16. **RC** - P15-7-10 At which depth die Fischer et al. observe similar patterns? It is also not clear, which location are you referring to. Therefore, the conclusion you make is not clear either.
    **AR** - Fischer et al., (2002) reported mass fluxes in sinking particles, which were collected in the following years after the collection of PF3 almost at the same location. Each mooring system (PF5, PF7, PF8) was deployed at two similar depth traps to the PF3. 'a location further to the south' is referred to site Bouvet Island (BO; 54.50° S, 3.33° W, Fischer et al., (2002)), which was located further to the south compared to site PF3. To make the sentence clear, we rephrased it.
    Author's changes in manuscript – 'Fischer et al., (2002) observed a similar seasonal flux pattern in the following years measured almost at the same depths and at the same location as site PF3. At site BO (54.50° S, 3.33° W), which was located further south than site PF3, the authors also found a period of almost no flux in July-December for four years approximately at 2200 m depth.'

17. **RC** - P16-30 This statement should be stronger (remove 'potentially'), because water T has an effect on GDGTs, and not the other way round.
    **AR** - We agree on it. 'potentially' is removed.

18. **RC** - P17-2 'Larger scatter towards colder temperatures ...'
    **AR** - Corrected.

19. **RC** - P17-4 '...relationship of maSSTs AND TEX86L values ...'
    **AR -** Corrected.

20. **RC** - P18-7 'similar range as'
    **AR** - Corrected.

21. **RC** - P18-13 'Warm biases AS with the ...'
    **AR** - Corrected.

22. **RC** - P19-16-17 '... or OH-GDGT-based calibrations ... the limitations of a single global TEX86L calibration…'
   **AR** - Corrected.

---

## Author Comment (AC2) · 13 Apr 2019

Dear editor,

We thank the two anonymous referees for evaluating our manuscript. We are pleased to learn that both referees have found our work interesting. Their comments and suggestions helped us to improve the manuscript. Below are our replies to the referees' comments in blue. Page numbers mentioned here refer to the original manuscript published on Biogeosciences discussions.

**Anonymous Referee #2**

**Referee's comments (RC) -** The manuscript by Park et al. reports on seasonality of archaeal fluxes and GDGT- based thermometry in sinking particles based on two case studies in high latitudes. The study is based on material collected in sediment traps at different depths. This approach is complementary to the collection of surface sediments and offers the opportunity to study processes and mechanisms lying to the signal acquisition in the sediments. An interesting point is made on depths of production of OH-GDGTs. and the consequences on RI-OH thermometers. To conclude this manuscript address important issues in the paleo-proxy community and the new set of data presented is interesting. I therefore recommend the publication of this manuscript with minor revisions detailed below:

**General comments**

1. **Referee's comments (RC)** - A more throughout presentation of the errors associated with the temperature reconstruction based on the different indices and different calibrations should be discussed and provided.

   **Author's responses (AR)** - Global $TEX_{86}$ calibrations are based on an assumption that major GDGT producers 'Thaumarchaeota' dwell mostly in surface waters and experience similar biogeochemical alterations crossed the global ocean. This greatly simplifies a diversity of ocean system. Next to analytical errors, seasonal and/or depth production of GDGTs or additional contribution of other archaeal community other than Thaumarchaeota can account for the calibration errors (Kim et al., 2010).

   The $TEX_{86}^{L}$ calibration has a 4.0 °C of uncertainty (standard error of the estimate), which could be even larger than a magnitude of annual temperature variability in cold oceans (e.g. our two regions). As appeared in the eastern Fram Strait, the changes of $TEX_{86}^{L}$-derived temperature are largely controlled by the depth and time of GDGT production and sinking materials aggregated with GDGTs by time. Absolute estimated temperatures varied within the calibration error. In this case, the temperature error inherited from the calibration is less important than other relative changes. In the Polar Front, warm biases observed at deep traps were larger than the calibration errors, suggesting significant non-thermal effects on GDGT compositions or the unreliability of the global calibration in the region.

   To assess the analytical error, we analyzed a lab-internal sediment standard. The standard deviation of replicate analyses is ±0.01 units of $TEX_{86}^{L}$ and 6 % for isoprenoidal GDGT concentrations.

   Author's changes in manuscript: We will add more statements regarding the differences between observed and reconstructed SST in the context of the analytical uncertainty and the calibration error as follow:

   (1) P2-Line13: 'Moreover, all $TEX_{86}^{L}$ calibrations for temperature include a rather large scatter, resulting in a calibration error of, e.g., ±4 °C for the $TEX_{86}^{L}$ calibration (Kim et al., 2010).' (2) Analytical error: The standard deviation of replicate analyses is reported in the section 'GDGT analyses'. (3) Calibration errors are given in the GDGT flux and indices section. (4) P11-L20: 'When the error of the $TEX_{86}^{L}$ calibration (±4 °C) is considered, the SST estimates are identical to the satellite-derived SSTs.' (4) P12-Line11: 'Warm and cold biases of the $TEX_{86}^{L}$-derived temperatures varied within the calibration error (±4 °C) throughout the trap deployment period. It shows that the bias of the calibration occurs neither in one direction only nor to the same extent even at a given location, instead the temperature estimate is more affected by other processes discussed above.' (5) P14-Line16: 'In the eastern Fram Strait, the changes of $TEX_{86}^{L}$-derived temperature are largely controlled by the depth and time of GDGT production and sinking materials aggregated with GDGTs by time. Additionally, the absolute estimated temperatures varied within the $TEX_{86}^{L}$ calibration error (±4 °C). In this case, the temperature error inherited from the calibration is

less important than other relative changes.' (6) P15-Line23: 'Temperature residuals (~7 °C) in the deep trap, which are larger than the calibration error (±4 ℃), suggest significant non-thermal effects on GDGT compositions or the unreliability of the global calibration in this region.'

2. **RC** - Different processes of the production as well as the export of GDGTs are investigated in in two settings, even if the figures are already numerous, it would be interesting to provide the reader with a figure/sketch summarizing the mechanisms of production (seasonality/community or depth changes) and export (type of ballasts or timing) in the two settings.

AR - We agree to have a figure or sketch summarizing all our findings. However, as you also noticed there are already 9 figures in the manuscript. We therefore chose to add a table (Table 2) rather than a figure. Additionally, we changed the format of Table1 to help the readers, who might want to compare Table1 to Table 2.

Author's changes in manuscript: Table 1. is restructured. New Table 2 is inserted in the summary and discussion section. Tables can be found below:

**Table 1.** Information on FEVI16 and PF3 trap.

| Trap name | FEVI16 | PF3 |
|---|---|---|
| Region | Eastern Fram Strait | Antarctic Polar Front |
| Location | | |
|    Latitude (° N) | 79.02 | -50.13 |
|    Longitude (° E) | 4.35 | 5.83 |
| Water depth (m) | 2580 | 3785 |
| Trap depth (m) | 1296 | 614 |
| | | 3196 |
| Deployment period (dd.mm.yyyy) | | |
| Start | 23.07.2007 | 10.11.1989 |
| End | 30.06.2008 | 23.12.1990 |
| Sampling interval (d) | 10-31 | 21, 42* |
| Cruise reports | ARK-XXII/1c (Klages and Participants, 2007) | ANT-VIII/3 (Gersonde and Participants, 1990) |
| | ARK-XXIII/2 (Kattner and Participants, 2009) | ANT-IX/2 (Fahrbach and Cruise Participants, 1992) |

*The exact sampling interval of each sample at FEVI and PF3 can be found on PANGAEA (https://doi.pangaea.de/10.1594/PANGAEA.897268).

**Table 2. Summary of $TEX_{86}^{L}$ thermometry in FEVI16 and PF3 site.**

| Trap name | FEVI16 | PF3 |
|---|---|---|
| Oceanographic setting | Seasonal ice cover | Winter ice edge |
| Main GDGT producers | Thaumarchaeota | Thaumarchaeota + Euryarchaeota |
| Surface ocean temperature | Satellite-SST[a]: -0.1 − 3.4 °C
Ave. SST[b]: 1.9 °C | Satellite-SST[a]: 1.8 − 5.2 °C
Ave. SST[b]: 3.5 °C |
| Shallow trap | $TEX_{86}^{L}$[c] T: 2.8 °C (30 − 80 m depth signal) | $TEX_{86}^{L}$[c] T: 4.6 °C (Thaumarchaeota + Euryarchaeota) |
| Deep trap | n.a. | $TEX_{86}^{L}$[c] T: 8.5 °C (Dominant Euryarchaeota) |
| Surface sediment | $TEX_{86}^{L}$ T:2.3/2.8 at 2400 m | $TEX_{86}^{L}$ T: 9.1°C at 3800 m |
| Relevant processes for GDGTs | - Export of upper ocean signal by fast settling particles
- Highly ballasted with opal and carbonate
- s.v.[d]: 15 m d$^{-1}$ | - Contribution of Euryarchaeota in CDW[e] casing warm biases |
| Conclusions | - Linear calibration ($TEX_{86}^{L}$) applicable
- Temporal offset due to changing ballast materials and s.v.
- OH-GDGTs based calibrations applicable | - Linear calibration ($TEX_{86}^{L}$) unreliable
- Nonlinear relationship between $TEX_{86}^{L}$ and SST (>50° N)
- OH-GDGTs based calibrations applicable |

[a]Satellite-derived sea surface temperature
[b]Averaged surface temperature over the trap deployment period
[c]Flux-weighted average temperature over the trap deployment period
[d]s.v.: sinking velocity
[e]CDW: Circumpolar Deep Water